# DiffiT: Diffusion Vision Transformers for Image Generation

## Abstract

Diffusion models with their powerful expressivity and high sample quality have enabled many new applications and use-cases in various domains. For sample generation, these models rely on a denoising neural network that generates images by iterative denoising. Yet, the role of denoising network architecture is not well-studied with most efforts relying on convolutional residual U-Nets. In this paper, we study the effectiveness of vision transformers in diffusion-based generative learning. Specifically, we propose a new model, denoted as Diffusion Vision Transformers (DiffiT), which consists of a hybrid hierarchical architecture with a U-shaped encoder and decoder. We introduce a novel time-dependent self-attention module that allows attention layers to adapt their behavior at different stages of the denoising process in an efficient manner. We also introduce latent DiffiT which consists of transformer model with the proposed self-attention layers, for high-resolution image generation. Our results show that DiffiT is surprisingly effective in generating high-fidelity images, and it achieves state-of-the-art (SOTA) benchmarks on a variety of class-conditional and unconditional synthesis tasks. In the latent space, DiffiT achieves a new SOTA FID score of **1.73** on **ImageNet-256** dataset. The code and pretrained models will be publicly available.

## 1 Introduction

Diffusion models (Sohl-Dickstein et al., 2015; Ho et al., 2020; Song et al., 2021b) have revolutionized the domain of generative learning, with successful frameworks in the front line such as DALL·E 3 (Ramesh et al., 2022), Imagen (Ho et al., 2022), Stable diffusion (Rombach et al., 2022; Rombach & Esser, 2022), and eDiff-I (Balaji et al., 2022). They have enabled generating diverse complex scenes in high fidelity which were once considered out of reach for prior models. Specifically, synthesis in diffusion models is formulated as an iterative process in which random image-shaped Gaussian noise is denoised gradually towards realistic samples (Sohl-Dickstein et al., 2015; Ho et al., 2020; Song et al., 2021b).

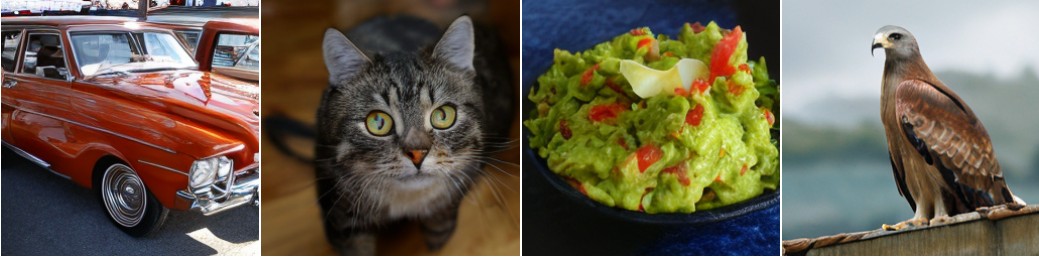

**Figure 1** – Uncurated 256×256 images generated by latent DiffiT, trained on ImageNet-256 Deng et al. (2009).

The core building block in this process is a *denoising autoencoder network* that takes a noisy image and predicts the denoising direction, equivalent to *the score function* (Vincent, 2011; Hyvärinen, 2005). This network, which is shared across different time steps of the denoising process, is often a variant of U-Net (Ronneberger et al., 2015; Ho et al., 2020) that consists of convolutional residual blocks as well as self-attention layers in several resolutions of the network. Although the self-attention

layers have shown to be important for capturing long-range spatial dependencies, yet there exists a lack of standard design patterns on how to incorporate them. In fact, most denoising networks often leverage self-attention layers only in their low-resolution feature maps (Dhariwal & Nichol, 2021) to avoid their expensive computational complexity.

Recently, several works (Balaji et al., 2022; Kreis et al., 2022; Choi et al., 2022) have observed that diffusion models exhibit a unique temporal dynamic during generation. At the beginning of the denoising process, when the image contains strong Gaussian noise, the high-frequency content of the image is completely perturbed, and the denoising network primarily focuses on predicting the low-frequency content. However, towards the end of denoising, in which most of the image structure is generated, the network tends to focus on predicting high-frequency details.

The time dependency of the denoising network is often implemented via simple temporal positional embeddings that are fed to different residual blocks via arithmetic operations such as spatial addition. In fact, the convolutional filters in the denoising network are not time-dependent and the time embedding only applies a channel-wise shift and scaling. Hence, such a simple mechanism may not be able to optimally capture the time dependency of the network during the entire denoising process.

In this work, we aim to address the following limitations: i) lack of self-attention design patterns in denoising networks ii) fine-grained control over capturing the time-dependent component. We introduce a novel Vision Transformer-based model for image generation, called DiffiT (pronounced *di-feet*) which achieves state-of-the-art performance in terms of FID score of image generation on CIFAR10 (Krizhevsky et al., 2009), AFHQv2-64 (Choi et al., 2020), and FFHQ-64 (Karras et al., 2019) (image space) as well as ImageNet-256 (Deng et al., 2009) (latent space) datasets.

Specifically, DiffiT proposes a new paradigm in which temporal dependency is only integrated into the self-attention layers where the key, query, and value weights are adapted per time step. This allows the denoising model to dynamically change its attention mechanism for different denoising stages. Our proposed self-attention leverages a window-based scheme without cross-communication among the local regions. This design is surprisingly effective and significantly reduces the expensive computational cost of self-attention. In an effort to unify the architecture design patterns, we also propose a hierarchical transformer-based architecture for latent space synthesis tasks.

The following summarizes our contributions in this work:

- We introduce a novel time-dependent self-attention module that is specifically tailored to capture both short- and long-range spatial dependencies. Our proposed time-dependent self-attention dynamically adapts its behavior over sampling time steps.
- We propose a novel hierarchical transformer-based architecture, denoted as DiffiT, which unifies the design patterns of denoising networks
- We show that DiffiT can achieve state-of-the-art performance on a variety of datasets for both image and latent space generation tasks.

## 2 METHODOLOGY

### 2.1 DIFFUSION MODEL

Diffusion models (Sohl-Dickstein et al., 2015; Ho et al., 2020; Song et al., 2021b) are a family of generative models that synthesize samples via an iterative denoising process. Given a data distribution as $q_0(\mathbf{z}_0)$, a family of random variables $\mathbf{z}_t$ for $t \in [0, T]$ are defined by injecting Gaussian noise to $\mathbf{z}_0$, *i.e.*, $q_t(\mathbf{z}_t) = \int q(\mathbf{z}_t|\mathbf{z}_0)q_0(\mathbf{z}_0)\mathrm{d}\mathbf{z}_0$, where $q(\mathbf{z}_t|\mathbf{z}_0) = \mathcal{N}(\mathbf{z}_0, \sigma_t^2 \boldsymbol{I})$ is a Gaussian distribution. Typically, $\sigma_t$ is chosen as a non-decreasing sequence such that $\sigma_0 = 0$ and $\sigma_T$ being much larger than the data variance. This is called the "Variance-Exploding" noising schedule in the literature (Song et al., 2021b); for simplicity, we use these notations throughout the paper, but we note that it can be equivalently converted to other commonly used schedules (such as "Variance-Preserving" (Ho et al., 2020)) by simply rescaling the data with a scaling term, dependent on $t$ (Song et al., 2021a; Karras et al., 2022).

The distributions of these random variables are the marginal distributions of forward diffusion processes (Markovian or not (Song et al., 2021a)) that gradually reduces the "signal-to-noise" ratio between the data and noise. As a generative model, diffusion models are trained to approximate the

reverse diffusion process, that is, to transform from the initial noisy distribution (that is approximately Gaussian) to a distribution that is close to the data one.

**Training** Despite being derived from different perspectives, diffusion models can generally be written as learning the following denoising autoencoder objective (Vincent, 2011)

$$\mathbb{E}_{q_0(\mathbf{z}_0), t \sim p(t), \epsilon \sim \mathcal{N}(0, \boldsymbol{I})}[\lambda(t) \| \epsilon - \epsilon_\theta(\mathbf{z}_0 + \sigma_t \epsilon, t)\|_2^2]. \quad (1)$$

Intuitively, given a noisy sample from $q(\mathbf{z}_t)$ (generated via $\mathbf{z}_t := \mathbf{z}_0 + \sigma_t \epsilon$), a neural network $\epsilon_\theta$ is trained to predict the amount of noise added (*i.e.*, $\epsilon$). Equivalently, the neural network can also be trained to predict $\mathbf{z}_0$ instead (Ho et al., 2020; Salimans & Ho, 2022). The above objective is also known as denoising score matching (Vincent, 2011), where the goal is to try to fit the data score (*i.e.*, $\nabla_{\mathbf{z}_t} \log q(\mathbf{z}_t)$) with a neural network, also known as the score network $s_\theta(\mathbf{z}_t, t)$. The score network can be related to $\epsilon_\theta$ via the relationship $s_\theta(\mathbf{z}_t, t) := -\epsilon_\theta(\mathbf{z}_t, t)/\sigma_t$.

**Sampling** Samples from the diffusion model can be simulated by the following family of stochastic differential equations that solve from $t = T$ to $t = 0$ (Grenander & Miller, 1994; Karras et al., 2022; Zhang et al., 2022; Dockhorn et al., 2021):

$$d\mathbf{z} = -(\dot{\sigma}_t + \beta_t)\sigma_t s_\theta(\mathbf{z}, t)dt + \sqrt{2\beta_t}\sigma_t d\omega_t, \quad (2)$$

where $\omega_t$ is the reverse standard Wiener process, and $\beta_t$ is a function that describes the amount of stochastic noise during the sampling process. If $\beta_t = 0$ for all $t$, then the process becomes a probabilistic ordinary differential equation (Anderson, 1982) (ODE), and can be solved by ODE integrators such as denoising diffusion implicit models (DDIM (Song et al., 2021a)). Otherwise, solvers for stochastic differential equations (SDE) can be used, including the one for the original denoising diffusion probabilistic models (DDPM (Ho et al., 2020)). Typically, ODE solvers can converge to high-quality samples in fewer steps and SDE solvers are more robust to inaccurate score models (Karras et al., 2022).

## 2.2 DIFFIT MODEL

**Time-dependent Self-Attention** At every layer, our transformer block receives $\{\mathbf{x_s}\}$, a set of tokens arranged spatially on a 2D grid in its input. It also receives $\mathbf{x_t}$, a time token representing the time step. Similar to Ho et al. (2020), we obtain the time token by feeding positional time embeddings to a small MLP with swish activation (Elfwing et al., 2018). This time token is passed to all layers in our denoising network. We introduce our time-dependent multi-head self-attention, which captures both long-range spatial and temporal dependencies by projecting feature and time token embeddings in a shared space. Specifically, time-dependent queries $\mathbf{q}$, keys $\mathbf{k}$ and values $\mathbf{v}$ in the shared space are computed by a linear projection of spatial and time embeddings $\mathbf{x_s}$ and $\mathbf{x_t}$ via

$$\mathbf{q_s} = \mathbf{x_s}\mathbf{W}_{qs} + \mathbf{x_t}\mathbf{W}_{qt}, \quad (3)$$
$$\mathbf{k_s} = \mathbf{x_s}\mathbf{W}_{ks} + \mathbf{x_t}\mathbf{W}_{kt}, \quad (4)$$
$$\mathbf{v_s} = \mathbf{x_s}\mathbf{W}_{vs} + \mathbf{x_t}\mathbf{W}_{vt}, \quad (5)$$

DiffiT Transformer Block

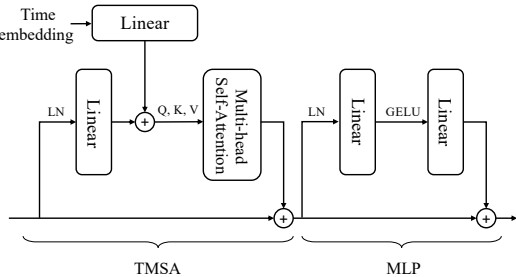

**Figure 2** – The DiffiT Transformer block applies linear projection to spatial and time-embedding tokens before combining them together to form query, key, and value vectors for each token. These vectors are then used to compute multi-head self-attention activations, followed by two linear layers. Above, LN indicates Layer Norm Ba et al. (2016) and GELU denotes the Gaussian error linear unit activation function Hendrycks & Gimpel (2016). TMSA (time-dependent multi-head self-attention) and MLP (multi-layer perceptron) are discussed in Eq. 7 and Eq. 8.

where $\mathbf{W}_{qs}, \mathbf{W}_{qt}, \mathbf{W}_{ks}, \mathbf{W}_{kt}, \mathbf{W}_{vs}, \mathbf{W}_{vt}$ denote spatial and temporal linear projection weights for their corresponding queries, keys, and values respectively.

We note that the operations listed in Eq. 3 to 5 are equivalent to a linear projection of each spatial token, concatenated with the time token. As a result, key, query, and value are all linear functions of both time and spatial tokens and they can adaptively modify the behavior of attention for different

time steps. We define $\mathbf{Q} := \{\mathbf{q_s}\}$, $\mathbf{K} := \{\mathbf{k_s}\}$, and $\mathbf{V} := \{\mathbf{v_s}\}$ which are stacked form of query, key, and values in rows of a matrix. The self-attention is then computed as follows

$$\text{Attention}(\mathbf{Q}, \mathbf{K}, \mathbf{V}) = \text{Softmax}\left(\frac{\mathbf{Q}\mathbf{K}^\top}{\sqrt{d}} + \mathbf{B}\right)\mathbf{V}. \tag{6}$$

In which, $d$ is a scaling factor for keys $\mathbf{K}$, and $\mathbf{B}$ corresponds to a relative position bias (Shaw et al., 2018). For computing the attention, the relative position bias allows for the encoding of information across each attention head. Note that although the relative position bias is implicitly affected by the input time embedding, directly integrating it with this component may result in sub-optimal performance as it needs to capture both spatial and temporal information. Please see Sec. 4.3 for more analysis.

**DiffiT Block** The DiffiT transformer block (see Fig. 2) is a core building block of the proposed architecture and is defined as

$$\hat{\mathbf{x}}_\mathbf{s} = \text{TMSA}\left(\text{LN}\left(\mathbf{x_s}\right), \mathbf{x_t}\right) + \mathbf{x_s}, \tag{7}$$
$$\mathbf{x_s} = \text{MLP}\left(\text{LN}\left(\hat{\mathbf{x}}_\mathbf{s}\right)\right) + \hat{\mathbf{x}}_\mathbf{s}, \tag{8}$$

where TMSA denotes time-dependent multi-head self-attention, as described in the above, $\mathbf{x_t}$ is the time-embedding token, $\mathbf{x_s}$ is a spatial token, and LN and MLP denote Layer Norm (Ba et al., 2016) and multi-layer perceptron (MLP) respectively.

Next, we describe how we design DiffiT models in image and latent space.

### 2.2.1 IMAGE SPACE

**DiffiT Architecture** DiffiT uses a symmetrical U-Shaped encoder-decoder architecture in which the contracting and expanding paths are connected to each other via skip connections at every resolution. Specifically, each resolution of the encoder or decoder paths consists of $L$ consecutive DiffiT blocks, containing our proposed time-dependent self-attention modules. In the beginning of each path, for both the encoder and decoder, a convolutional layer is employed to match the number of feature maps. In addition, a convolutional upsampling or downsampling layer is also used for transitioning between each resolution. we speculate that the use of these convolutional layers embeds inductive image bias that can further improve the performance. In the remainder of this section, we discuss the DiffiT Transformer block and our proposed time-dependent self-attention mechanism. We use our proposed Transformer block as the residual cells when constructing the U-shaped denoising architecture.

**Window Attention** The quadratic cost of attention scales poorly when the number of spatial tokens is large, especially in the case of large feature maps. Without loss of generality, the above Transformer block can be applied to local regions, in which the self-attention is computed within non-overlapping partitioned windows. Although these partitioned windows do not allow information to be propagated between different regions, the U-Net structure with bottleneck layers permits information sharing between different regions.

**DiffiT ResBlock** We define our final residual cell by combining our proposed DiffiT Transformer block with an additional convolutional layer in the form:

$$\hat{\mathbf{x}}_\mathbf{s} = \text{Conv}_{3\times3}\left(\text{Swish}\left(\text{GN}\left(\mathbf{x_s}\right)\right)\right), \tag{9}$$
$$\mathbf{x_s} = \text{DiffiT-Block}\left(\hat{\mathbf{x}}_\mathbf{s}, \mathbf{x_t}\right) + \mathbf{x_s}, \tag{10}$$

where GN denotes the group normalization operation (Wu & He, 2018) and DiffiT-Transformer is defined in Eq. 7 and Eq. 8 (shown in Fig. 2). Our residual cell for image space diffusion models is a hybrid cell combining both a convolutional layer and our Transformer block.

### 2.2.2 LATENT SPACE

Recently, latent diffusion models have been shown effective in generating high-quality large-resolution images (Vahdat et al., 2021; Rombach et al., 2022). We also extend DiffiT to latent diffusion models. For this, we first encode the images using a pre-trained variational auto-encoder network (Rombach et al., 2022). The feature maps are then converted into non-overlapping patches and projected into a

**Table 1** – FID performance comparison against various generative approaches on the CIFAR10, FFHQ-64 and AFHQv2-64 datasets. VP and VE denote Variance Preserving and Variance Exploding respectively. DiffiT outperforms competing approaches, sometimes by large margins.

| Method | Class | Space Type | CIFAR-10 32×32 | FFHQ 64×64 | AFHQv2 64×64 |
|---|---|---|---|---|---|
| NVAE (Vahdat & Kautz, 2020) | VAE | - | 23.50 | - | - |
| GenViT (Yang et al., 2022) | Diffusion | Image | 20.20 | - | - |
| AutoGAN (Gong et al., 2019) | GAN | - | 12.40 | - | - |
| TransGAN (Jiang et al., 2021) | GAN | - | 9.26 | - | - |
| INDM (Kim et al., 2022) | Diffusion | Latent | 3.09 | - | - |
| DDPM++ (VE) (Song et al., 2021b) | Diffusion | Image | 3.77 | 25.95 | 18.52 |
| U-ViT (Bao et al., 2022) | Diffusion | Image | 3.11 | - | - |
| DDPM++ (VP) (Song et al., 2021b) | Diffusion | Image | 3.01 | 3.39 | 2.58 |
| StyleGAN2 w/ ADA (Karras et al., 2020) | GAN | - | 2.92 | - | - |
| LSGM (Vahdat et al., 2021) | Diffusion | Latent | 2.01 | - | - |
| EDM (VE) (Karras et al., 2022) | Diffusion | Image | 2.01 | 2.53 | 2.16 |
| EDM (VP) (Karras et al., 2022) | Diffusion | Image | 1.99 | 2.39 | 1.96 |
| **DiffiT** (Ours) | Diffusion | Image | **1.95** | **2.22** | **1.87** |

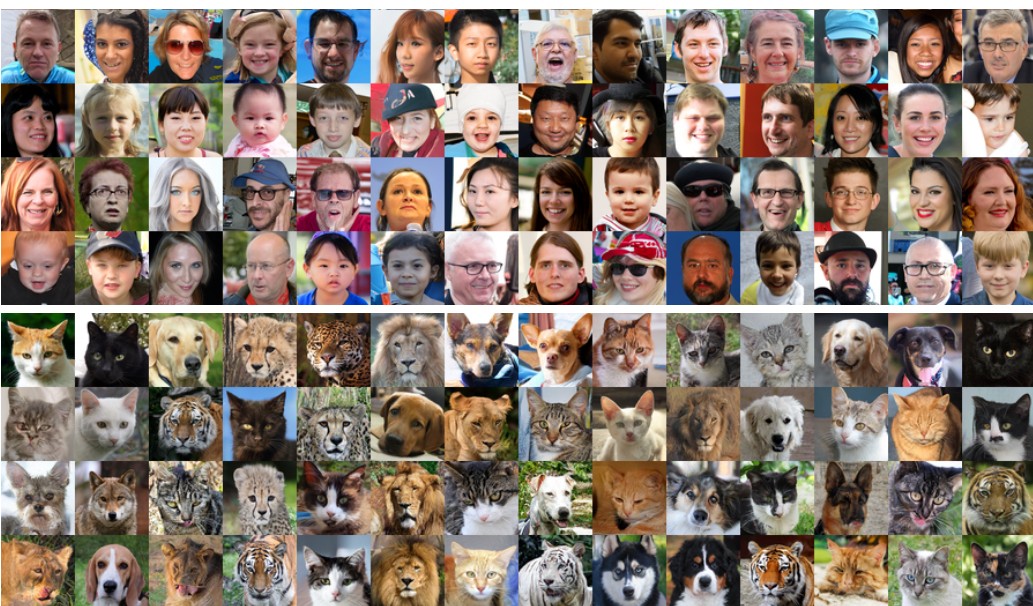

**Figure 3** – Visualization of uncurated generated images for FFHQ-64 and AFHQv2-64 datasets. Images are randomly sampled. Best viewed in color.

new embedding space. Similar to the DiT model (Peebles & Xie, 2022), we use a vision transformer, without upsampling or downsampling layers, as the denoising network in the latent space. In addition, we also utilize a three-channel classifier-free guidance to improve the quality of generated samples. The final layer of the architecture is a simple linear layer to decode the output. Unlike DiT (Peebles & Xie, 2022), our model does not use additional adaLN layers to incorporate the time dependency, as it leverages the proposed TMSA blocks for this purpose. Please see the supplementary materials for more details on the latent DiffiT architecture as well as the training setting.

## 3 RESULTS

### 3.1 IMAGE SPACE EXPERIMENTS

We have trained the proposed DiffiT model on three datasets of CIFAR-10, FFHQ and AFHQv2 respectively. In Table. 1, we compare the performance of our model against a variety of different generative models including other score-based diffusion models as well as GANs, and VAEs. DiffiT achieves a state-of-the-art image generation FID score of 1.95 on the CIFAR-10 dataset, outperforming

**Table 2** – Comparison of image generation performance against state-of-the-art models on ImageNet-256 dataset. Synthesized images have a resolution of 256×256. The latent DiffiT model achieves SOTA performance in terms of FID score and demonstrates a competitive performance across other metrics.

| Method | Class | FID | sFID | IS | Precision | Recall |
|---|---|---|---|---|---|---|
| ADM (Dhariwal & Nichol, 2021) | Diffusion | 10.94 | 6.02 | 100.98 | 0.69 | **0.63** |
| LDM-4 (Rombach et al., 2022) | Diffusion | 10.56 | - | 103.49 | 0.71 | 0.62 |
| BigGAN-deep (Brock et al., 2018) | GAN | 6.95 | 7.36 | 171.40 | **0.87** | 0.28 |
| MaskGIT (Chang et al., 2022) | Autoregressive | 4.02 | - | **355.60** | - | - |
| RQ-Transformer (Lee et al., 2022) | Autoregressive | 3.80 | - | 323.70 | - | - |
| ADM-G (Dhariwal & Nichol, 2021) | Diffusion | 3.94 | 6.14 | 215.84 | 0.83 | 0.53 |
| LDM-4-G (Rombach et al., 2022) | Diffusion | 3.60 | - | 247.67 | **0.87** | 0.48 |
| StyleGAN-XL (Sauer et al., 2022) | GAN | 2.30 | **4.02** | 265.12 | 0.78 | 0.53 |
| DiT-XL/2-G (Karras et al., 2022) | Diffusion | 2.27 | 4.60 | 278.24 | 0.83 | 0.57 |
| MDT (Gao et al., 2023) | Diffusion | 1.79 | 4.57 | 283.01 | 0.81 | 0.61 |
| **DiffiT** (Ours) | Diffusion | **1.73** | 4.54 | 276.49 | 0.80 | 0.62 |

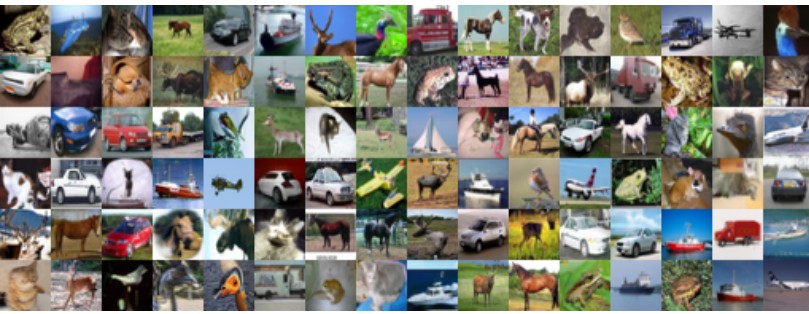

**Figure 4** – Visualization of uncurated generated images for CIFAR-10 dataset. Best viewed in color.

state-of-the-art diffusion models such as EDM (Karras et al., 2022) and LSGM (Vahdat et al., 2021). In comparison to two recent ViT-based diffusion models, our proposed DiffiT significantly outperforms U-ViT (Bao et al., 2022) and GenViT (Yang et al., 2022) models in terms of FID score in CIFAR-10 dataset.

In Table 1, we also present quantitative benchmarks for image generation performance on FFHQ-64 and AFHQv2-64. For both FFHQ-64 and AFHQv2-64, DiffiT significantly outperforms EDM (Karras et al., 2022) and DDPM++ (Song et al., 2021b) models, both on VP and VE training configurations, in terms of FID score. We illustrate the generated images from DiffiT models trained on FFHQ-64/AFHQv2-64 and CIFAR-10 datasets in Fig. 3 and Fig. 4 respectively.

### 3.2 LATENT SPACE EXPERIMENTS

We have also trained the latent DiffiT model for images with the 256×256 resolution on the ImageNet-256 dataset. In Table. 1, we present a comparison against other approaches using various image quality metrics. For this comparison, we select the best performance metrics from each model which may include techniques such classifier-free guidance. The latent DiffiT model outperforms competing approaches, such as DiT-XL/2-G and StyleGAN-XL, in terms of FID score and sets a new state-of-the-art performance. In terms of other metrics such as IS and sFID, the latent DiffiT model shows a competitive performance, hence indicating the effectiveness of the proposed time-dependant self-attention. In addition, in Fig. 5, we show a visualization of uncurated images that are generated on ImageNet-256 dataset. We observe that latent DiffiT model is capable of generating diverse high quality images across different classes.

### 4 ABLATION

In this section, we provide additional ablation studies to provide insights into DiffiT. We address three main questions: (1) How do different components of DiffiT contribute to the final generative performance, (2) What is the optimal way of introducing time dependency in our Transformer block? and (3) How does our time-dependent attention behave as a function of time?

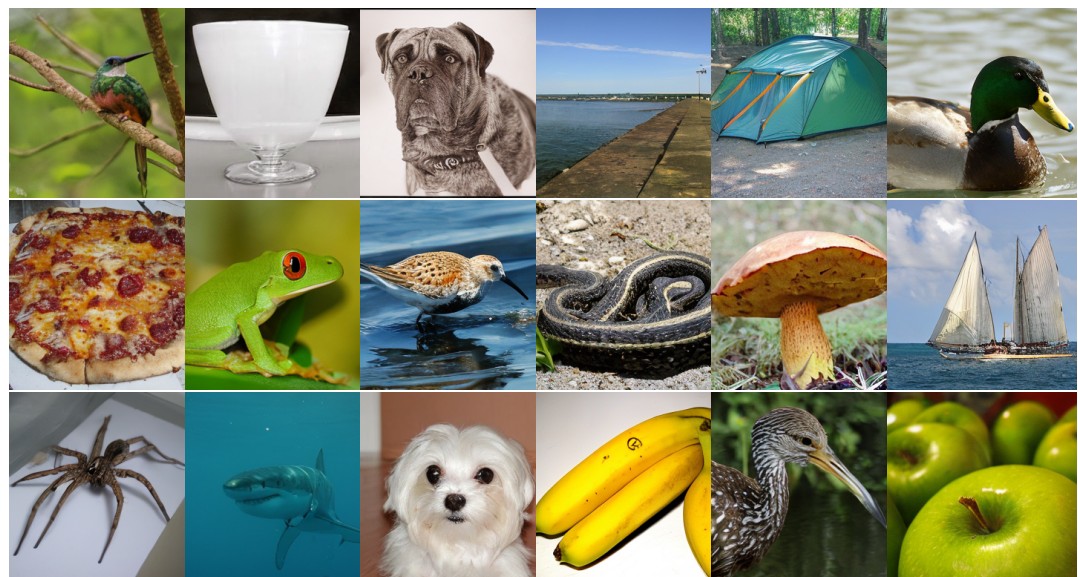

**Figure 5** – Visualization of uncurated generated 256×256 images on ImageNet-256 (Deng et al., 2009) dataset by latent DiffiT model. Images are randomly sampled. Best viewed in color.

## 4.1 EFFECT OF ARCHITECTURE DESIGN

As presented in Table 3, we study the effect of various components of both encoder and decoder in the architecture design on the image generation performance in terms of FID score on CIFAR-10.

For these experiments, the projected temporal component is adaptively scaled and simply added to the spatial component in each stage. We start from the original ViT (Dosovitskiy et al., 2020) base model with 12 layers and employ it as the encoder (config A). For the decoder, we use the Multi-Level Feature Aggregation variant of SETR (Zheng et al., 2021)

**Table 3** – Effectiveness of encoder and decoder in DiffiT.

| Config | Encoder | Decoder | FID Score |
|---|---|---|---|
| A | ViT (Dosovitskiy et al., 2020) | SETR-MLA (Zheng et al., 2021) | 5.34 |
| B | + Multi-Resolution | SETR-MLA (Zheng et al., 2021) | 4.64 |
| C | Multi-Resolution | + Multi-Resolution | 3.71 |
| D | + DiffiT Encoder | Multi-Resolution | 2.27 |
| E | + DiffiT Encoder | + DiffiT Decoder | **1.95** |

(SETR-MLA) to generate images in the input resolution. Our experiments show this architecture is sub-optimal as it yields a final FID score of 5.34. We hypothesize this could be due to the isotropic architecture of ViT which does not allow learning representations at multiple scales.

We then extend the encoder ViT into 4 different multi-resolution stages with a convolutional layer in between each stage for downsampling (config B). We also employ a window-based approach for computing self-attention in a local manner and use the same window sizes as described in Sec. A. We denote this setup as Multi-Resolution and observe that these changes and learning multi-scale feature representations in the encoder substantially improve the FID score to 4.64.

In addition, instead of SETR-MLA (Zheng et al., 2021) decoder, we construct a symmetric U-like architecture by using the same Multi-Resolution setup except for using convolutional layer between stage for upsampling (config C). These changes further improve the FID score to 3.71. Furthermore, we first add the DiffiT Transformer blocks and construct a DiffiT Encoder and observe that FID scores substantially improve to 2.27 (config D). As a

**Table 4** – Effectiveness of TMSA.

| Model | FID |
|---|---|
| DDPM++(VE) (Song et al., 2021b) | 3.77 |
| DDPM++(VE) w/ TMSA | **3.49** |
| DDPM++(VP) (Song et al., 2021b) | 3.01 |
| DDPM++(VP) w/ TMSA | **2.76** |

result, this validates the effectiveness of the proposed TMSA in which the self-attention models both spatial and temporal dependencies. Using the DiffiT decoder further improves the FID score to 1.95 (config E), hence demonstrating the importance of DiffiT Transformer blocks for decoding.

## 4.2 TIME-DEPENDANT SELF-ATTENTION

We evaluate the effectiveness of our proposed TMSA layers in a generic denoising network. Specifically, using the DDPM++ (Song et al., 2021b) model, we replace the original self-attention layers with TMSA layers for both VE and VP settings for image generation on the CIFAR10 dataset. Note that we did not change the original hyper-parameters for this study. As shown in Table 4 employing TMSA decreases the FID scores by 0.28 and 0.25 for VE and VP settings respectively. These results demonstrate the effectiveness of the proposed TMSA to dynamically adapt to different sampling steps and capture temporal information.

## 4.3 IMPACT OF SELF-ATTENTION COMPONENTS

In Table 5, we study different design choices for introducing time-dependency in self-attention layers. In the first baseline, we remove the temporal component from our proposed TMSA and we only add the temporal tokens to relative positional bias (config F). We observe a significant increase in the FID score to 3.97 from 1.95. In the second

**Table 5** – The impact of self-attention components.

| Config | Component | FID Score |
|--------|-----------|-----------|
| F | Relative Position Bias | 3.97 |
| G | MLP | 3.81 |
| H | TMSA | **1.95** |

baseline, instead of using relative positional bias, we add temporal tokens to the MLP layer of DiffiT Transformer block (config G). We observe that the FID score slightly improves to 3.81, but it is still sub-optimal compared to our proposed TMSA (config H). Hence, this experiment validates the effectiveness of our proposed TMSA that integrates time tokens directly with spatial tokens when forming queries, keys, and values in self-attention layers.

## 4.4 VISUALIZATION OF SELF-ATTENTION MAPS

One of our key motivations in defining time-dependent self-attention is to allow the self-attention module to adapt its behavior dynamically for different stages of the denoising process. To demonstrate that our introduced self-attention layer achieves this, in Fig. 6, we visualize attention maps from a token at the center of a feature map to all $32 \times 32$ tokens around it during the sampling trajectory of a model trained on CIFAR-10. We observe that at the early stages of sampling the token attends to most other tokens around it. Towards the end of the generation process, attention maps mostly follow the object mask.

## 4.5 MODEL PARAMETER COMPARISONS

We study the significance of model capacity in generating high-quality images by comparing the number of parameters in Table 6. All models use the same number of function evaluations for sample generation for fair comparisons. In image and latent space experiments, our comparison includes FFHQ-64 and ImageNet-256 dataset with best competing models. For FFHQ-64 dataset, we observe that DiffiT outperforms EDM (Karras et al., 2022) in terms of FID score by a large margin, despite having only %1.31 more parameters. Furthermore, for ImageNet-256 dataset, despite outperforming DiT-XL/2-G (Karras et al., 2022) in terms of FID score, the latent DiffiT model has %12.59 less number of parameters. The substantial difference in the number of parameters is due to using the tailored TMSA blocks instead of AdaLN layers. Hence, the proposed design could yield more parameter-efficient models while maintaining better or comparable image generation quality.

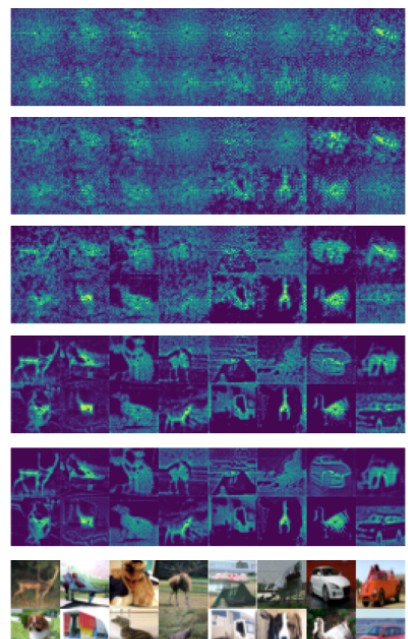

**Figure 6** – Visualization of the attention maps in TMSA block for a token at the center of a feature map along sampling trajectory. The corresponding generated images are shown in the bottom. In early stages of sampling, the token attends to most tokens around it. Towards the end of the generation, attention maps follow the object mask.

## 4.6 EFFECT OF CLASSIFIER-FREE GUIDANCE

We investigate the effect of classifier-free guidance scale on the quality of generated samples in terms of FID score. As shown in Fig. 4.6, we observe that increasing the scale up to guidance value of 5 improves the FID score. However, further increase of the guidance scale decreases the quality of generated images.

**Table 6** – Comparison of Model Parameters.

| Model | Space | Parameters (M) | FID |
|---|---|---|---|
| EDM (Karras et al., 2022) | Image | 75 | 2.39 |
| **DiffiT** | Image | 76 | **2.22** |
| DiT-XL/2-G (Karras et al., 2022) | Latent | 675 | 2.27 |
| **DiffiT** | Latent | 590 | **2.20** |

## 5 RELATED WORK

Diffusion models (Sohl-Dickstein et al., 2015; Ho et al., 2020; Song et al., 2021b) have driven significant advances in generative learning in various domains such as text-to-image generation (Ramesh et al., 2022; Saharia et al., 2022; Balaji et al., 2022). With the introduction of vision transformers (Dosovitskiy et al., 2020) and their success in different tasks such as image recognition tasks (*e.g.* classification), there have been several efforts (Luhman & Luhman, 2022; Bao et al., 2022; Peebles & Xie, 2022) to harness their strengths for diffusion image generation. The image-space models such as U-ViT (Bao et al., 2022) have not yet achieved superior performance over state-of-the-art ap-

**Figure 7** – Effect of classifier-free guidance on FID.

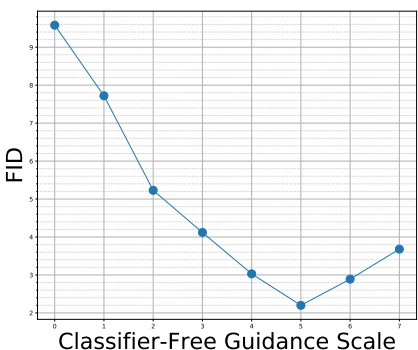

proaches and their success remains limited. For latent diffusion models however, DiT (Peebles & Xie, 2022) has achieved state-of-the-art performance for high-resolution image generation on ImageNet-1K dataset.

The proposed DiffiT has a different design compared to U-ViT (Bao et al., 2022). U-ViT (Bao et al., 2022) uses an isotropic vision transformer encoder with spatial self-attention and a linear layer as the decoder. However, DiffiT has a multi-resolution architecture with a tailored window-based time-dependent self-attention. Furthermore, the DiT (Peebles & Xie, 2022) uses Adaptive LayerNorm (AdaLN) for capturing the time-dependent component into various network layers. As opposed to this design, the latent DiffiT model uses the proposed time-dependant self-attention layers for this purpose and does not employ AdaLN in its architecture. According to our benchmarks, DiffiT outperforms both U-ViT (Bao et al., 2022) and DiT (Peebles & Xie, 2022) models on CIFAR10 and ImageNet-256 datasets in terms of FID of generated samples.

## 6 CONCLUSION

In this work, we presented Diffusion Vision Transformers (DiffiT) which is a hybrid transformer-based model for diffusion-based image generation. The proposed DiffiT model unifies the design pattern of denoising diffusion architectures. Specifically, we proposed a novel time-dependent self-attention layer that jointly learns both spatial and temporal dependencies. As opposed to previous efforts, our proposed self-attention allows for selectively capturing both short and long-range information in different time steps. Analysis of time-dependent self-attention maps reveals strong localization and dynamic temporal behavior over sampling steps. In addition, we introduced the latent DiffiT model for high-resolution image generation.

We have evaluated the effectiveness of DiffiT using both image and latent space experiments. DiffiT achieves state-of-the-art performance in terms of FID score on the CIFAR10, FFHQ-64, and AFHQv2-64 datasets, sometimes by large margins. The latent DiffiT model also achieves state-of-the-art FID scores on the ImageNet-256 dataset.

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

# APPENDIX

In this supplementary materials. We first provide the implementation details in Sec. A. We then discuss the the detailed image and latent space architectures of the DiffiT model in Sec. B along with exact details of all operations. Furthermore, in Sec. C, we qualitative results of generated images from all datasets including FFHQ-64 (Karras et al., 2019), AFHQv2-64 (Choi et al., 2020), CIFAR10 (Krizhevsky et al., 2009) and ImageNet-256 (Deng et al., 2009) datasets.

## A  IMPLEMENTATION DETAILS

### A.1  IMAGE SPACE

For fair comparison, we strictly followed the training configurations and data augmentation strategies of the EDM (Karras et al., 2022) model for the experiments on CIFAR10 (Krizhevsky et al., 2009), FFHQ-64 (Karras et al., 2019) and AFHQv2-64 (Choi et al., 2020) datasets, all in an unconditional setting. All the experiments were trained for 200000 iterations with Adam optimizer (Kingma & Ba, 2014) and used PyTorch 1.12.1 framework and 8 NVIDIA A100 GPUs. We used batch sizes of 512, 256 and 256 and learning rates of $1 \times 10^{-3}$, $2 \times 10^{-4}$ and $2 \times 10^{-4}$ and training images of sizes $32 \times 32$, $64 \times 64$ and $64 \times 64$ on experiments for CIFAR10 (Krizhevsky et al., 2009), FFHQ-64 (Karras et al., 2019) and AFHQv2-64 (Choi et al., 2020) datasets, respectively.

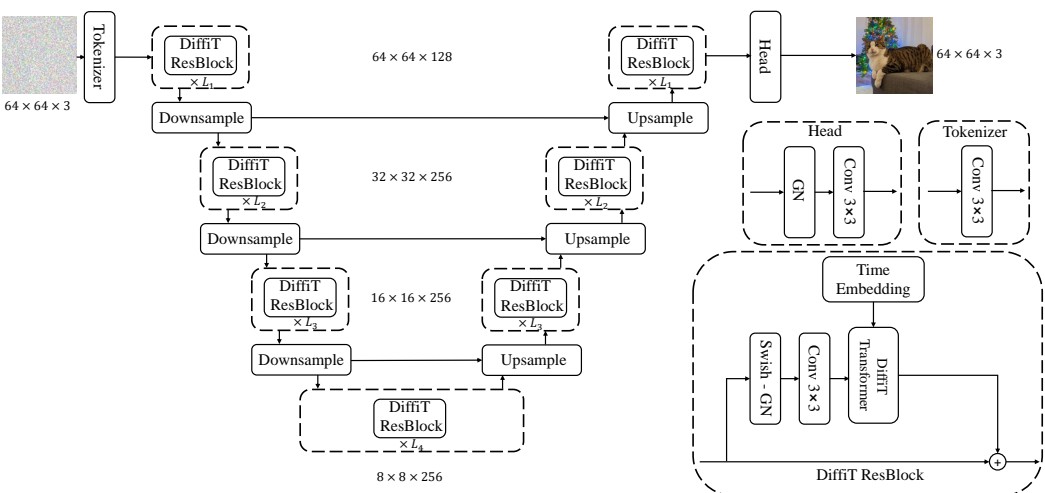

**Figure S.1** – Overview of the DiffiT framework. Downsample and Upsample denote convolutional downsampling and upsampling layers, respectively. Please see the main paper for detailed information regarding the architecture and DiffiT Transformer blocks.

We use the deterministic sampler of EDM (Karras et al., 2022) model with 18, 40 and 40 steps for CIFAR-10, FFHQ-64 and AFHQv2-64 datasets respectively. For FFHQ-64 and AFHQv2-64 datasets, our DiffiT network spans across 4 different stages with 1, 2, 2, 2 blocks at each stage. We also use window-based attention with local window size of 8 at each stage. For CIFAR-10 dataset, the DiffiT network has 3 stages with 2 blocks at each stage. Similarly, we compute attentions on local windows with size 4 at each stage. Note that for all networks, the resolution is decreased by a factor of 2 in between stages. However, except for when transitioning from the first to second stage, we keep the number of channels constant in the rest of the stages to maintain both the number of parameters and latency in our network.

As also discussed in Sec. 2.2, without loss of generality and to provide a fair comparison across different networks, we employ traditional convolutional-based downsampling and upsampling layers for transitioning into lower or higher resolutions. We achieved similar image generation performance by using bilinear interpolation for feature resizing instead of convolution. Furthermore, for fair comparison, in all of our experiments, we used the FID score which is computed on 50K samples and using the training set as the reference set.

**Table S.1** – Detailed description of components in DiffiT encoder for models that are trained at $64 \times 64$ resolution.

| Component Description | Output size |
| --- | --- |
| Input | $64 \times 64 \times 3$ |
| Tokenizer | $64 \times 64 \times 128$ |
| DiffiT ResBlock $\times L_1$ | $64 \times 64 \times 128$ |
| Downsampler | $32 \times 32 \times 128$ |
| DiffiT ResBlock $\times L_2$ | $32 \times 32 \times 256$ |
| Downsampler | $16 \times 16 \times 128$ |
| DiffiT ResBlock $\times L_3$ | $16 \times 16 \times 256$ |
| Downsampler | $8 \times 8 \times 256$ |
| DiffiT ResBlock $\times L_4$ | $8 \times 8 \times 256$ |

**Table S.2** – Detailed description of components in DiffiT decoder for models that are trained at $64 \times 64$ resolution.

| Component Description | Output size |
| --- | --- |
| Input | $8 \times 8 \times 256$ |
| Upsampler | $16 \times 16 \times 256$ |
| DiffiT ResBlock $\times L_3$ | $16 \times 16 \times 256$ |
| Upsampler | $32 \times 32 \times 256$ |
| DiffiT ResBlock $\times L_2$ | $32 \times 32 \times 256$ |
| Upsampler | $64 \times 64 \times 256$ |
| DiffiT ResBlock $\times L_1$ | $64 \times 64 \times 128$ |
| Head | $64 \times 64 \times 3$ |

### A.2 LATENT SPACE

In addition, we employ a learning rate of $3 \times 10^{-4}$ and a batch size of 256 and without weight decay. We also use the exponential moving average (EMA) of weights using a decay of 0.9999. We also use the same diffusion hyper-parameters as in the ADM (Dhariwal & Nichol, 2021) model. Our best results are achieved by using a classifier-free guidance scale of 5.0. The model is trained using 32 NVIDIA A100 GPU for 6M iterations. For a fair comparison, we use the DDPM (Ho et al., 2020) sampler with 250 steps and report FID-50K for ImageNet-256 (Deng et al., 2009) experiments.

## B ARCHITECTURE

### B.1 IMAGE SPACE

We present the architecture of DiffiT network in Fig. S.1. In addition, we provide the details of blocks and their corresponding output sizes for both the encoder and decoder of the DiffiT model in Table S.1 and Table S.2, respectively. The presented architecture details denote models that are trained with $64{\times}64$ resolution. Without loss of generality, the architecture can be extended for $32{\times}32$ resolution. For FFHQ-64 (Karras et al., 2019) and AFHQv2-64 (Choi et al., 2020) datasets, the values of $L_1$, $L_2$, $L_3$ and $L_4$ are 4, 4, 4, and 4 respectively. For CIFAR-10 (Krizhevsky et al., 2009) dataset, the architecture spans across three different resolution levels (*i.e.* 32, 16, 8), and the values of $L_1$, $L_2$, $L_3$ are 4, 4, 4 respectively. Please refer to the paper for more information regarding the architecture details.

### B.2 LATENT SPACE

For fair comparison against DiT (Peebles & Xie, 2022), we use the same settings for ViT architecture. This includes the number of layers, self attention heads, etc. Our model is comparable to DiT-XL/2-G variant which uses a patch size of 2. Specifically, we use a depth of 30 layers with hidden size dimension of 1152, number of heads dimension of 16 and MLP ratio of 4.

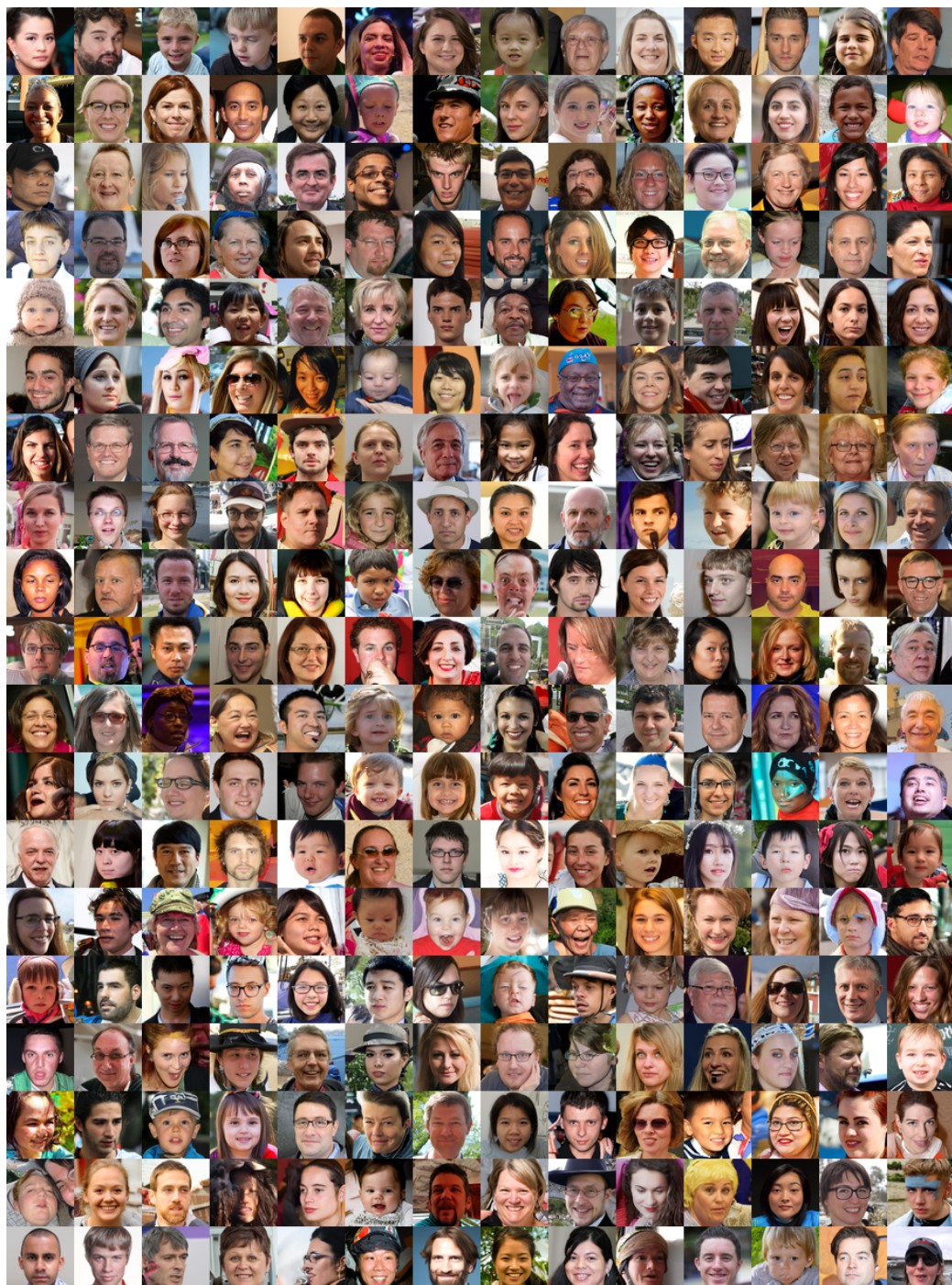

**Figure S.2** – Visualization of uncurated generated images for FFHQ-64 (Karras et al., 2019) dataset. Best viewed in color.

## C    QUALITATIVE RESULTS

In Figures S.2, S.3, S.4, S.5 and S.6, we visualize additional samples generated on the FFHQ-64 (Karras et al., 2019), AFHQv2-64 (Choi et al., 2020), CIFAR-10 (Krizhevsky et al., 2009) and ImageNet-256 (Deng et al., 2009) datasets, respectively. Evidently, the proposed DiffiT model is capable of capturing fine-grained details and produce high fidelity images across all datasets.

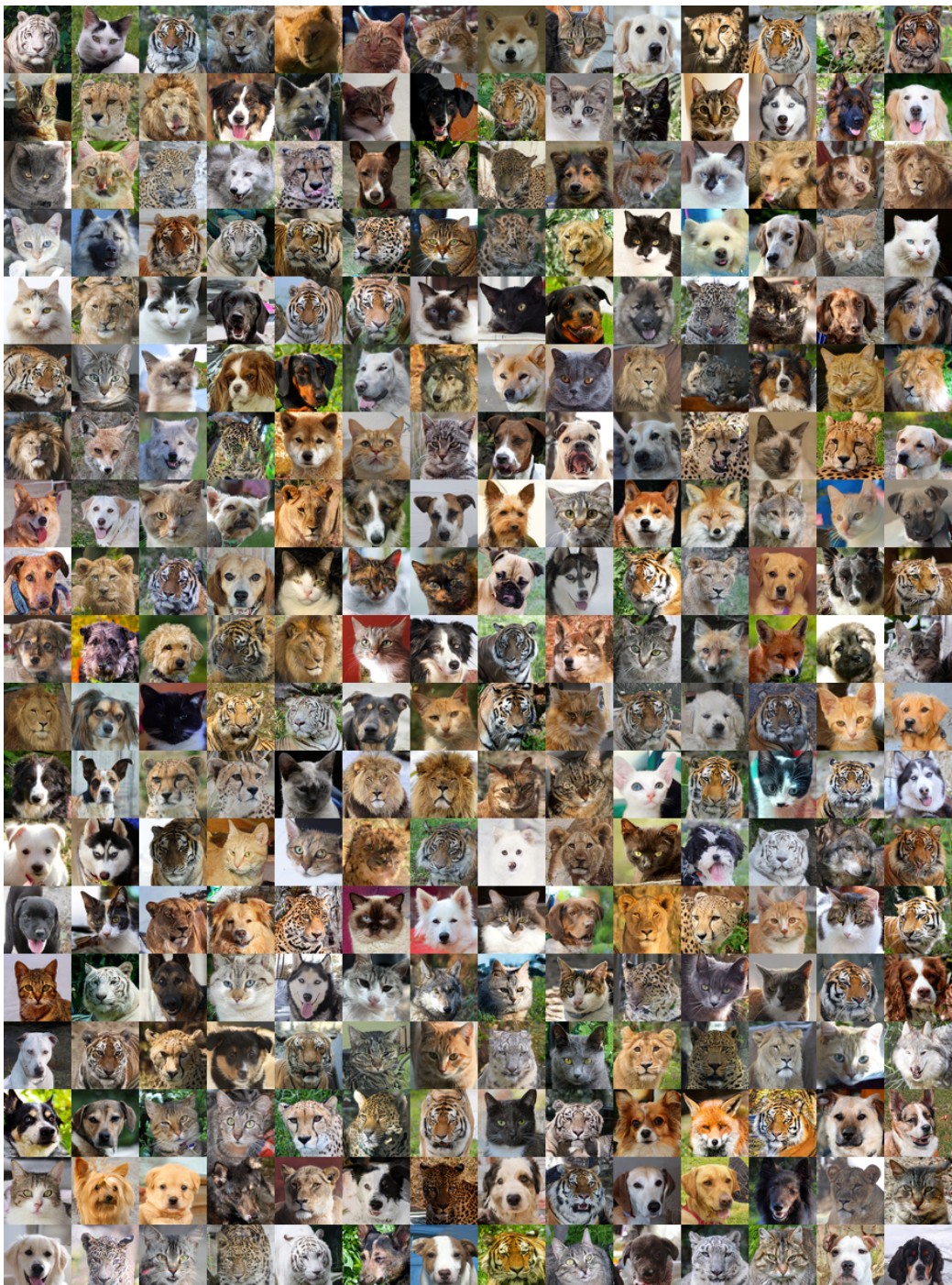

**Figure S.3** – Visualization of uncurated generated images for AFHQv2-64 (Choi et al., 2020) dataset. Best viewed in color.

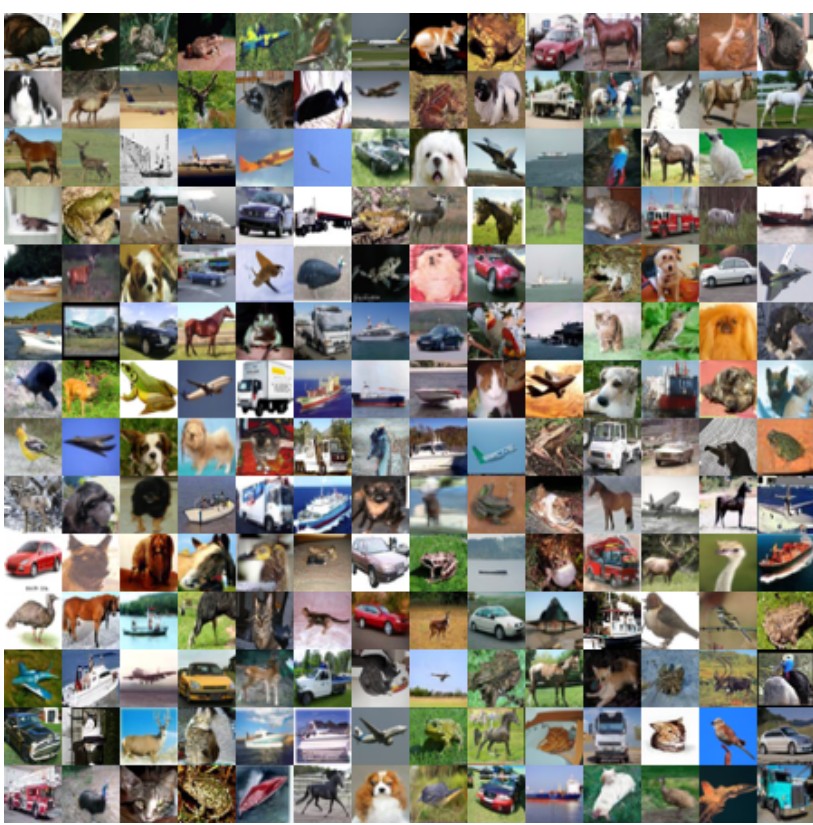

**Figure S.4** – Visualization of uncurated generated images for CIFAR-10 (Krizhevsky et al., 2009) dataset. Best viewed in color.

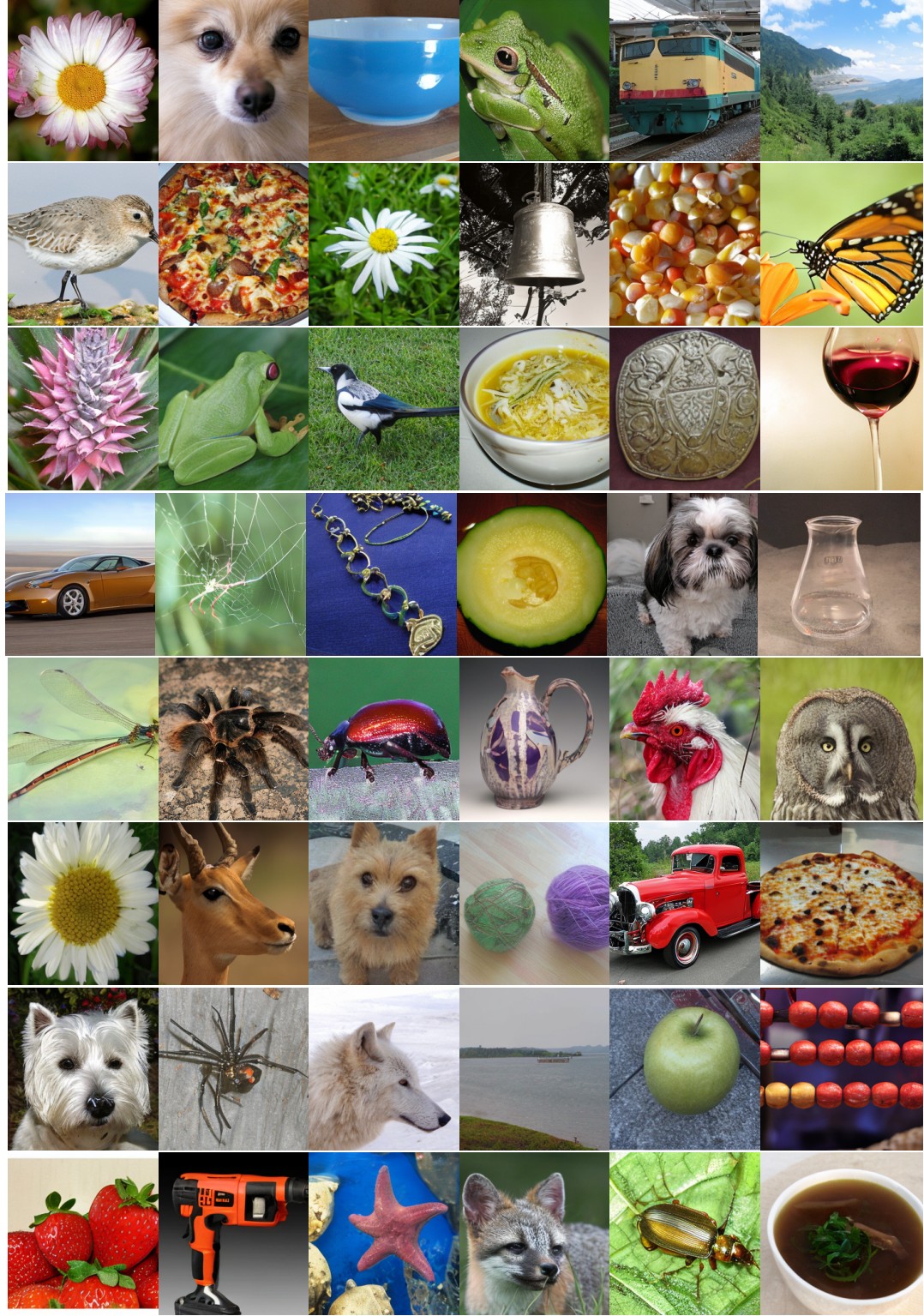

**Figure S.5** – Visualization of uncurated generated 256×256 images on ImageNet-256 (Deng et al., 2009) dataset by latent DiffiT model. Images are randomly sampled. Best viewed in color.

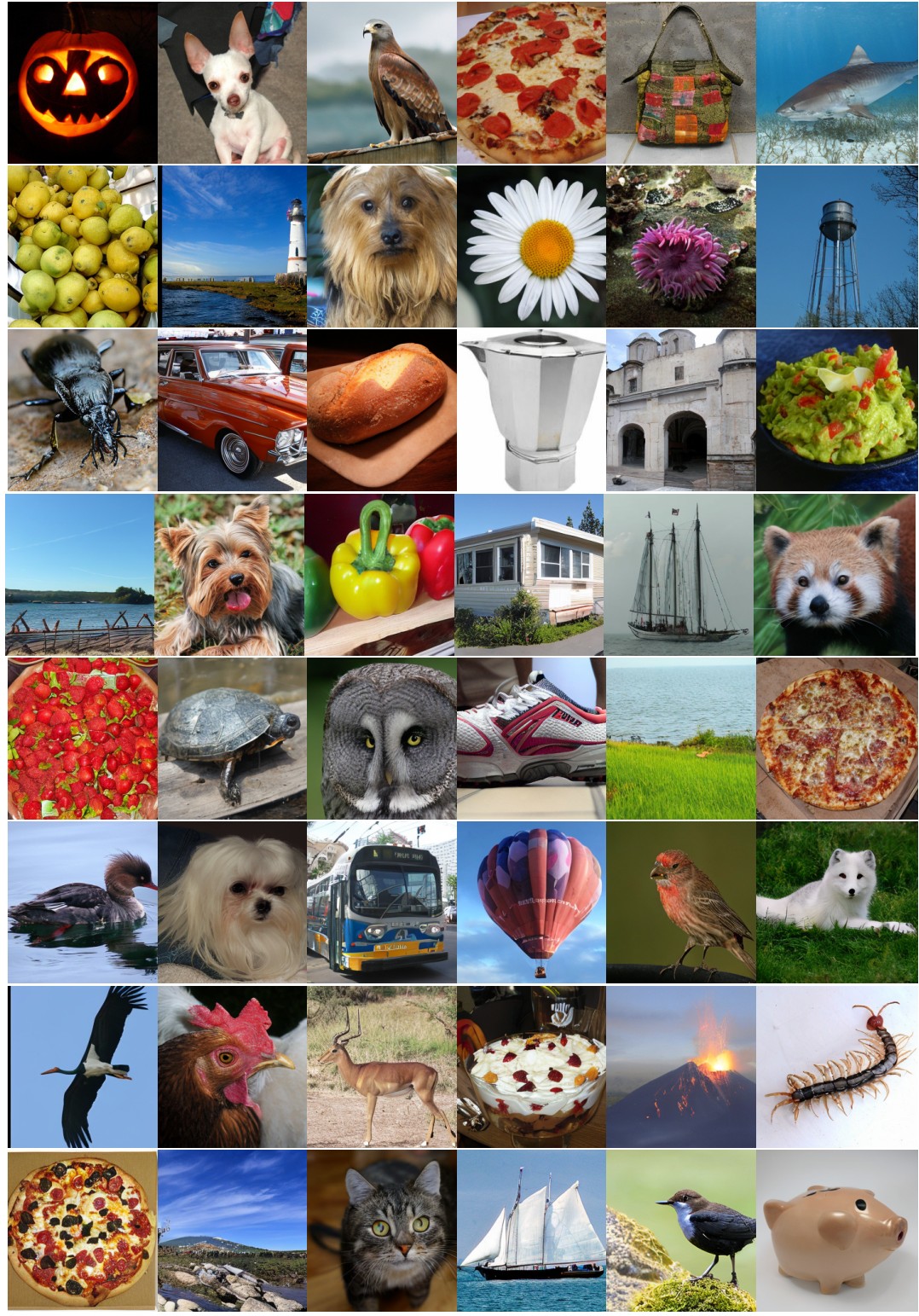

**Figure S.6** – Visualization of uncurated generated 256×256 images on ImageNet-256 (Deng et al., 2009) dataset by latent DiffiT model. Images are randomly sampled. Best viewed in color.

