# OpenReview forum: "DiffiT: Diffusion Vision Transformers for Image Generation"
_ICLR.cc/2024/Conference — ICLR 2024 Conference Withdrawn Submission_

### Official Review · Reviewer_w9cP · 2023-10-28

**Soundness:** 3 good
**Presentation:** 3 good
**Contribution:** 3 good
**Rating:** 5
**Confidence:** 5

**Summary:**

This article investigates the application of Vision Transformer in diffusion models and proposes a new method to embed time step information in Self-Attention layer, allowing the attention layer to have different behaviors based on different time steps. The effectiveness of the proposed method is evaluated in pixel space and latent space, achieving state-of-the-art FID on ImageNet-256.

**Strengths:**

1. A new method of injecting time step information into the Self-Attention layer.
2. A new multi-scale Transformer structure for diffusion models.
3. The effectiveness was verified on multiple datasets and achieved state-of-the-art FID.

**Weaknesses:**

1. The proposed method in the article mainly improves the embedding of temporal information. Can this method be applied to other information? For example, can it be applied to injecting other sequential information such as text?

2. The results of DiffiT in Table 2 and Table 6 are different. It is necessary to add a column to show model size in Table 2. Also in Table 1. Besides, Eencoder and decoder params need to be separately added to Table 3.

3. In the experiments of Table 3, default timestep injection is scaled and added to spatial component. Are there other literature that does this? Here, a fair comparison needs to be made with DiT's AdaLN and U-ViT's concat as input token methods while keeping other structures and hyper-parameters the same.

4. Window-based attention is also performed on resolutions of 32 and 64. What would happen if it is not used? Is window-based attention used for low resolution due to training resource limitations or because its effect is comparable to global attention? How does it compare with SwinTransformer?

**Questions:**

See the weaknesses.

---

### Official Review · Reviewer_okW1 · 2023-10-28

**Soundness:** 2 fair
**Presentation:** 2 fair
**Contribution:** 2 fair
**Rating:** 5
**Confidence:** 4

**Summary:**

This paper proposes a transformer-based diffusion backbone (DiffiT) with U-shaped encoder and decoder analogous to U-Net. It introduces a time-dependent self-attention module which could be composed as latent DiffiT for high-resolution image generation similar to DiT. DiffiT achieves SOTA results on CIFAR-10, FFHQ and AFHQv2,  and DiffiT achieves a new SOTA FID score of 1.73 (compared to previous SOTA 1.79) on ImageNet-256 generation.

**Strengths:**

1. The paper is well written with clear presentations.
2. The introduced time-dependent self-attention module is well motivated and shown to be effective with improved FID scores on generative tasks.

**Weaknesses:**

1. The main contribution of this paper is a new time modulation design for diffusion backbone, the underlying reasons why it should be more optimal than "shift and scaling" of standard residual blocks could be more thoroughly discussed and analyzed. Also it could be compared with the concept of "expert denoisers” as in [1].
2. The transformer-based architecture is highly analogous to U-Net which undermines the "novel" architecture claim.
3. The SOTA generation results on ImageNet-256 are not sufficiently solid: as shown in Tab.2 comparing to MDT the difference on FID is 1.73 vs 1.79, and the results under all the other metrics are not the best.

[1] eDiff-I: Text-to-Image Diffusion Models with Ensemble of Expert Denoisers

**Questions:**

1. For the ablation in Tab. 4, does the TSMA module introduce more learning parameters?
2. Could a similar visualization of Fig.6 be presented for the self-attention module in standard U-Net?

---

### Official Review · Reviewer_qgiP · 2023-10-31

**Soundness:** 2 fair
**Presentation:** 3 good
**Contribution:** 2 fair
**Rating:** 5
**Confidence:** 5

**Summary:**

This paper proposes a new network architecture for the diffusion model, which includes a U-shaped encoder and decoder, time-dependent self-attention, and other designs such as window attention and ResBlock. This network achieves state-of-the-art performance in image generation across multiple benchmarks.

**Strengths:**

- The overall writing is clear, making it easy to grasp the key design of the network.
- The time-dependent self-attention module is straightforward and easy to understand.
- The proposed network achieves state-of-the-art performance on multiple image generation benchmarks.

**Weaknesses:**

- The overall novelty of this work is limited because it primarily combines several existing modules to enhance performance, such as the U-Net-like structure, window attention, and local conv blocks. Could you clarify the novelty of your designs?
- The time-dependent self-attention is intriguing. I concur that the time step information is crucial for the diffusion model. However, I'm curious about the distinction between this design and the AdaLN used by DiT. Introducing time step information into the input of self-attention or the kqv might not yield a significant difference, I suspect. Moreover, there are no experiments to substantiate this claim.
- In Fig. 6, the attention maps vary across different time steps. I'm curious whether other networks like DiT and U-vit possess similar attention maps. This could serve as crucial evidence supporting the efficacy of the proposed time-dependent self-attention.

**Questions:**

- Clarify the novelty of the proposed network designs compared to existing ones.
- Verify the effeteness and mechanism of the proposed time-dependent self-attention.

**Details Of Ethics Concerns:**

No ethics concerns in this paper.

---

### Official Review · Reviewer_rYjo · 2023-11-01

**Soundness:** 3 good
**Presentation:** 3 good
**Contribution:** 2 fair
**Rating:** 5
**Confidence:** 5

**Summary:**

This paper proposed a new diffusion model, denoted as Diffusion Vision Transformers (DiffiT) for image generation. The core innovation is time-dependent self-attention module, which is designed to address temporal dynamic problem.

**Strengths:**

This paper is well written, and extensive experiments demonstrate that the generated image quality is good

**Weaknesses:**

The motivation in Intro section is not very clear. The Intro say diffusion models exhibit a unique temporal dynamic during generation, but do not claim what the drawback of this problem, and why it is important to address this  problem. And there are no enough experiment results to demonstrate that the proposed method can address this problem. Most of results show the image quality, do not show that the temporal dynamic can be well addressed.

**Questions:**

I am a little confused about what the relationship between this phenomenon and the temporal step encodings. Can